# A Novel *Perilla frutescens* (L.) Britton Cell-Derived Phytocomplex Regulates Keratinocytes Inflammatory Cascade and Barrier Function and Preserves Vaginal Mucosal Integrity In Vivo

**DOI:** 10.3390/pharmaceutics15010240

**Published:** 2023-01-10

**Authors:** Giovanna Pressi, Giovanna Rigillo, Paolo Governa, Vittoria Borgonetti, Giulia Baini, Raffaella Rizzi, Chiara Guarnerio, Oriana Bertaiola, Marco Frigo, Matilde Merlin, Stefania Paltrinieri, Roberto Zambonin, Stefano Pandolfo, Marco Biagi

**Affiliations:** 1Aethera Biotech s.r.l., 36043 Camisano Vicentino, Italy; 2Department of Biomedical, Metabolic and Neural Science, University of Modena and Reggio Emilia, 41125 Modena, Italy; 3Department of Biotechnology, Chemistry and Pharmacy (Department of Excellence 2018–2022), University of Siena, 53100 Siena, Italy; 4Department of Neuroscience, Psychology, Drug Research and Child Health (NEUROFARBA), University of Florence, 50139 Florence, Italy; 5Department of Physical Sciences, Earth and Environment, University of Siena, 53100 Siena, Italy; 6Eurochem Ricerche s.r.l., 35035 Mestrino, Italy

**Keywords:** *Perilla frutescens*, rosmarinic acid, phytocomplex, skin integrity, inflammation

## Abstract

In the last years, the medicinal plant *Perilla frutescens* (L.) Britton has gained scientific interest because leaf extracts, due to the presence of rosmarinic acid and other polyphenols, have shown anti-allergic and skin protective potential in pre-clinical studies. Nevertheless, the lack of standardized extracts has limited clinical applications to date. In this work, for the first time, a standardized phytocomplex of *P. frutescens*, enriched in rosmarinic acid and total polyphenols, was produced through innovative in vitro cell culture biotechnology and tested. The activity of perilla was evaluated in an in vitro inflammatory model of human keratinocytes (HaCaT) by monitoring tight junctions, filaggrin, and loricrin protein levels, the release of pro-inflammatory cytokines and JNK MAPK signaling. In a practical health care application, the perilla biotechnological phytocomplex was tested in a multilayer model of vaginal mucosa, and then, in a preliminary clinical observation to explore its capacity to preserve vaginal mucosal integrity in women in peri-menopause. In keratinocytes cells, perilla phytocomplex demonstrated to exert a marked activity in epidermis barrier maintenance and anti-inflammatory effects, preserving tight junction expression and downregulating cytokines release through targeting JNK activation. Furthermore, perilla showed positive effects in retaining vaginal mucosal integrity in the reconstructed vaginal mucosa model and in vivo tests. Overall, our data suggest that the biotechnological *P. frutescens* phytocomplex could represent an innovative ingredient for dermatological applications.

## 1. Introduction

*Perilla frutescens* (L.) Britton, also known as Shiso or simply perilla, is an annual herbaceous species belonging to the Lamiaceae family, widely cultivated in Asia and used as an ornamental, aromatic and medicinal plant. *P. frutescens* has historical importance in Asian medicine, but it is known in modern phytotherapy mainly to treat seasonal allergy symptoms [1]. In recent years, this species has been investigated in vitro and animal models for the potential role of leaf extracts in atopic dermatitis [2] and UV-induced skin damage [3]. The biological effects of *P. frutescens* preparations have been linked to its peculiar polyphenolic phytocomplex in which rosmarinic acid (RA) has the main role [4].

RA in perilla preparations, in fact, has been claimed to exert different protective activities, mainly counteracting cell oxidative stress and inflammatory markers upregulation [5], over than contributing to anti-asthmatic effect, in association with apigenin [6]. 

As reported by Yan and co-authors [7], RA (up to 1.57%), besides perillaldheyde and perillaketone, could be properly considered as a chemical marker of the leaves of the species and their different preparations, even if a large variability has been observed in samples of different origin. To date, a monograph in Official Pharmacopoeias and in official chemical essays have not been released for *P. frutescens*; as a consequence of these important issues, and considering that perilla is mostly used in the sector of food supplements and cosmetics [8], the standardization of perilla preparations used for health purposes is far from being achieved. 

One method for obtaining contaminant-free, and chemically standardized herbal phytocomplexes in industrial quantities, is to use in vitro cell cultures; this technology allows for solving problems related to the variability of herbal products since it provides preparations with a controlled content of active substances and a high profile of reproducibility and standardization.

For about 10 years, scientific research has developed and investigated the technology of in vitro plant cell cultures to obtain biologically active natural products [9,10], but the current challenge is to produce highly standardized phytocomplexes that could provide the fundamental multitarget and synergistic effect of secondary metabolites that is at the basis of the rationale for the use of phytotherapy nowadays. 

Our research group recently published an investigation on the production and biological potential of some biotechnological phytocomplexes, such as *Rosa chinensis* Jacq. [11], *Melissa officinalis* L. [12] and *Rhus coriaria* L. [13].

In this work, for the first time, we report the production and standardization of a biotechnological *P. frutescens* phytocomplex (PFP), obtained from plant cell culture, by monitoring the RA and anthocyanidin content. 

The conceptualization in developing PFP has been the attempt to produce a chemically standardized product with sounding skin-protective characteristics, such as being a candidate for topical applications in counteracting ailments affecting the skin epithelium, including mucosae. Minor dermatological disorders are too often underrated and have important pharmacological limits that scientific research tries to face. Among these, vaginal mucosa atrophy represents one of the most characteristic consequences accompanying menopause, impacting the life quality of menopausal women [14,15,16]. 

The present work aimed to develop and evaluate the safety and effectiveness of the new biotechnological PFP by investigating its biological properties. To do that, we planned our study in a three-step experimental design.

In the first phase, we tested PFP activity on skin inflammation and barrier maintenance in human keratinocytes by investigating the cellular and molecular mechanisms underpinning the effect. In the second phase, we evaluated the irritant and hydrating effects of PFP in an in vitro reconstructed 3D vaginal mucosa; finally, in the third phase, according to the normative that permits cosmetic ingredients testing, we performed a preliminary clinical observation in 30 healthy volunteers in pre- or post-menopause to evaluate the vaginal hydration, elasticity and extensibility after the application of a cosmetic preparation containing PFP. 

## 2. Materials and Methods

### 2.1. Perilla Frutescens Cell Culture

In this study, plants of *Perilla frutescens* L. Britton were used as starting plant material to generate a cell culture and bought and certified from the nursery plant “Le Georgiche”, Brescia, Italy. The botanical species authentication of *P. frutescens* was ensured and confirmed through molecular biology analysis (DNA fingerprint) performed in collaboration with Padano Technology Park, Lodi, Italy [17]. The young leaves of *P. frutescens* were washed under running water and sterilized through a sequencing treatment in ethanol 70% (*v*/*v*) (Honeywell, Wunstorfer Straße 40, D-30926 Seelze, Germany), water for about 1 min, sodium hypochlorite solution 2% (*v*/*v*) (6–14% active chlorine, (Merck KGaA, Germany) and Tween-20 0.1% (*v*/*v*) (Duchefa, Postbus 809, 2003 RV-Haarlem, The Netherlands) for 2–3 min and, finally, at least 3 washes with sterile distilled water. The sanitized plant tissue has been cut into minute fragments (explants) and deposited in Petri dishes containing solidified nutrient medium Gamborg B5 [18] with different combinations and concentrations of plant growth regulators (2,4 dichlorophenoxyacetic acid with and without 6-benzylaminopurine, 2,4 dichlorophenoxyacetic acid with and without kinetin, naphthalene acetic acid with and without kinetin, indole 3-acetic acid with and without Kinetin, naphthalene acetic acid and indole 3-acetic acid with and without kinetin and Picloram with and without 6-benzylaminopurine) and incubated at 25 °C and in dark conditions. The highest rate of callus growth was observed using the solid medium Gamborg B5 supplemented with sucrose 20 g/L (Sudzucker), Plant Agar 0.9% (*w*/*v*) (Duchefa), naphthalene acetic acid (NAA) 0.5 mg/L (Duchefa), indole 3-acetic acid (IAA) 1 mg/L, and pH adjusted to 6.5 (perilla medium). Calli grown on perilla medium were subjected to subculture for at least 6 months until they became friable and homogeneous, with a constant growth rate (*P. frutescens* stable cell line). 

Cell suspension cultures were generated by transferring 10% (*w*/*v* of selected callus into 250 mL of liquid culture medium Gamborg B5 supplemented with sucrose 20 g/L (Sudzucker), NAA 0.5 mg/L (Duchefa) and IAA 1 mg/L (Duchefa). The pH was adjusted to 6.5 before autoclaving (perilla liquid medium). The suspension cultures were maintained in a climatic room, in dark conditions, at 25 °C on a rotary shaker in constant agitation at 120 rpm and were subcultured in a new liquid medium every 14 days of fermentation. Afterward, to produce large quantities of biomass, the suspension culture was transferred and adapted to growth in a bioreactor of progressively increasing size (3L and 5L volume) with an amount of cell suspension inoculated into the liquid medium equal to 12% *v*/*v*. To increase the content of rosmarinic acid (RA) and total anthocyanins, after 14 days of fermentation in perilla liquid medium, the cell suspension was transferred to a final liquid medium (Gamborg B5 with the addition of sucrose 50 g/L, NAA 0.3 mg/L, and IAA 0.8 mg/L). The pH was adjusted to 5.9 before autoclaving (perilla final liquid medium). The suspension culture was grown for a culture cycle of 21 days in a climatic room at 25 °C on a rotary shaker in constant agitation at 120 rpm and in dark conditions.

### 2.2. Phytocomplex Preparation from Perilla Frutescens Selected Cell Culture

After 21 days of growth in perilla final liquid medium, at 25 °C and in the dark, the *P. frutescens* cell suspension was filtered by a 50 µm mesh filter, and the medium cultures discarded. Cells were washed with twice the volume of saline solution (0.9% *w*/*v* NaCl in sterile water), added with citric acid 1.5% (*w*/*w*), and then homogenized with ultra-turrax at 15,000 rpm for 20 min. The biomass of homogenized cells was dried using a Mini Spray Dryer (BUCHI-B290) to obtain a powder of PFP with a high content of RA and anthocyanidins [19]. 

### 2.3. UPLC-DAD Analysis

100 mg of PFP powder was dissolved in 30 mL of ethanol/water (60:40 *v*/*v*) for the quantification of RA and total polyphenols expressed as equivalent in RA. For the quantification of the total anthocyanidins, 100 mg of PFP were dissolved in 30 mL of ethanol/water/12N hydrochloric acid (60:39:1 *v*/*v*/*v*). The suspension was mixed for 30 s and sonicated for 15 min in an ice bath; finally, it was centrifuged at 4000 rpm for 15 min at 6 °C. The supernatant was collected and diluted 1:10 (first 1:5 in a solvent and then 1:2 in water), then filtered over 0.22 μm filters. Five independent replicates of PFP were extracted and analyzed. The chromatography system and method used are described in [12]. Identification of RA was assessed by using reference standard, whereas the identification of other polyphenols in the sample was based on UV/VIS spectra of compounds with λ_max_ at 280 and 330 nm, characteristic of hydroxycinnamic derivates that occur in perilla. The amount of RA was evaluated through the interpolation of peak area at 330 nm from standard curve built with RA (purity ≥ 98%, code R4033, Merck KGaA, Germany); total polyphenols content (expressed as equivalent of RA) was calculated as the sum of peak areas at 330 nm of all compounds identified as above described.

Identification of anthocyanidins was performed by monitoring compounds with characteristic λ_max_ at 520 nm. The content of anthocyanidins (expressed as equivalent of cyanidine-3-O-glucoside) was evaluated through the interpolation of the sum peak areas at 520 nm of all compounds identified in the class from the calibration curve of cyanidine-3-O-glucoside (purity ≥ 96%; 0915S, Extrasynthese, Genay, France). The data analysis was carried out with Empower 3 software.

### 2.4. Cell Culture and Treatments

The human foreskin fibroblasts (HFF), human keratinocytes from adult skin (HaCaT) and human monocytes (THP-1) were cultured at 37 °C and 5% CO_2_ in DMEM or RPMI 1640 (Merck KGaA, Germany) supplemented with 10% heat-inactivated fetal bovine serum (FBS) (Merck KgaA, Germany), 1% penicillin/streptomycin solution (Merck KgaA, Germany), and 1% of L-glutamine (Merck KgaA, Germany) and passed by trypsinization. PFP stock solution was prepared by solubilizing the extract in the related cell culture medium. 

The inflammatory condition was induced by the bacterial lipopolysaccharide (LPS from Gram-negative *Salmonella enteritidis*, #L7770, Merck KgaA) alone or in association with the oxidative stimulus H_2_O_2_, as reported in each of the following sections. The control group received the related fresh or unconditioned cell medium. Cells were collected at time points indicated in each section for further analysis.

### 2.5. Cell Viability

Cell viability was tested using Cell Counting kit-8 (CCK-8, Merck KgaA, Germany) as previously described [20]. Briefly, HaCaT and HFF cells (5 × 10^4^) were cultured in 96-well plates. Treatments were performed by incubating cells with PFP (0.1, 1, 10, 100, 1000 µg/mL) for 24 h, then CCK-8 solution was added to each well and incubated at 37 °C, 5% CO_2_ for 60 min. Cell viability was calculated by measuring the absorbance at 450 nm.

### 2.6. Wound Healing

Wound healing assay was performed according to the published method [13]. Briefly, HaCaT cells (5 × 10^4^) were seeded into 6-well cell culture plates and allowed to grow to 70–80% confluence as a monolayer. The monolayer was gently scratched across the center of the well with a sterile one-mL pipette tip. A second scratch was performed in a perpendicular way to the first, creating a cross in each well. After scratching, the medium was removed, and the wells were washed twice in PBS (Merck KgaA, Germany) solution. Fresh medium containing 5% *v*/*v* of heat-inactivated FBS and PFP (100 µg/mL) was added to each well. Images were obtained from the same fields immediately after scratching (t_0_) and after 6 h and 24 h using a Leica DMIL microscope, and analyzed using ImageJ software by manually selecting the wound region and recording the total area.

The experiments were conducted In triplicate, and two fields were analyzed for each replicate (*n* = 6). Untreated scratched cells represented the control. The percentage of wound closure was calculated using the following formula:[(Wound area t_0_ − Wound area t)/Wound area t_0_] × 100

### 2.7. Pro-Collagene I and Elastin Dosage

HFF cells (5 × 10^3^) were seeded into 96-well plates and grown for 24 h. Cells were treated with PFP 100 µg/mL in DMEM 1% FBS for 24 h; then, supernatants were collected. Pro-collagen I and elastin dosage were measured by using non-competitive sandwich ELISA (Human Pro-Collagen I Iα ELISA kit (ab210966), Abcam and Human Elastin ELISA Kit (ab239433), Abcam, UK), according to the manufacturer’s instructions. Positive controls furnished by Abcam were used.

### 2.8. Cytokines Dosage

Pro-inflammatory cytokines (TNF-α, IL-6, and IL-1β) production was evaluated in the following conditions:

(1) THP-1 cells (5 × 10^4^ cells/well) were seeded in 75 cm^2^ flasks and stimulated with LPS 200 ng/mL for 24 h. HaCaT cells (5 × 10^4^ cells/well) were seeded into 24-well plates and, after growing for 24 h, were co-treated with PFP (1, 10, 100 µg/mL) and LPS-conditioned THP-1 medium (LCTM) for 24 h.

(2) HaCaT cells (5 × 10^4^ cells/well) were seeded into 24-well plates and, after growing for 24 h, were co-treated with PFP (100 µg/mL) and LPS (1 µg/mL) for 24 h.

(3) THP-1 cells (5 × 10^4^) were seeded and, after growing for 24 h, were stimulated with PFP (100 µg/mL) and LPS (200 ng/mL) for 24 h. Cytokines release was measured together in HaCaT cell lysate and medium by non-competitive sandwich ELISA (Biolegend, Thermo Fisher Scientific, MA, USA) according to the manufacturer’s instruction. Absorbance was measured at 450 nm using an MP96 microplate. Experimental conditions followed as described in our previous works [13,21].

### 2.9. Leukocytes Infiltration Test

The leukocyte infiltration test was performed as previously described [21], with some minor modifications. Briefly, HaCaT cells were seeded on coated slide and cultured until confluence. The cells were pre-treated with PFP (100 µg/mL) for 2 h and then exposed to LPS (1 µg/mL) for 24 h. Thereafter, THP-1 cells (3 × 10^5^) were added to the well and co-cultured with HaCaT cells for 1 h. After washing, the slides were stained with eosin-haematoxylin, and the images were captured by microscope.

### 2.10. MAPKs Activation

MAPKs (JNK) activation was evaluated using non-competitive sandwich ELISA (Biolegend, Thermo Fisher Scientific, MA, USA) according to the manufacturer’s instructions as previously described by [20]; positive control furnished by the kit producer was used to validate the test. 

HaCaT cells (1 × 10^5^) were seeded into 24-well plates and cultured for 24 h. Cells were co-treated with PFP (100 μg/mL) and LCTM for 60 min. The activation of JNK was calculated as the ratio of phosphorylated to total proteins, normalizing values to the untreated control. 

### 2.11. Immunofluorescence

HaCaT cells (5 × 10^4^) were seeded in 24-well plates and grown for 24 h. Cells were pre-treated for 2 h with PFP (100 µg/mL), then stimulated with LPS (250 µg/mL) and H_2_O_2_ (2mM) for 3 h. After, cells were fixed with 4% paraformaldehyde for 15 min at room temperature (RT). Following permeabilization with 0.5% PBS-Triton X-100 for 10 min, and incubation with blocking buffer (PBS, containing 1% BSA) for 1 h at RT, cells were incubated at 4 °C overnight with primary antibodies anti-ZO-1 (1:100, Genetex, CA, USA), anti-occludin (1:100, Genetex, CA, USA). Cells were washed and incubated with secondary antibodies labeled with Invitrogen Alexa Fluor 488 (Thermo Fisher Scientific, MA, USA) for 1 h at RT. Finally, staining with DAPI was performed for 10 min. The visualization and acquisition of images were performed under the microscope using EVOS fluorescence (Thermo Fisher Scientific, MA, USA).

### 2.12. Western Blot

HFF cells (1 × 10^5^) were seeded in 12-well plates and grown for 24 h. Cells were pre-treated for 2 h with PFP (100 µg/mL), then stimulated with LPS (250 µg/mL) and H_2_O_2_ (2mM) for 3 h. Cells were trypsinized, harvested and lysed in Laemmly buffer (25 mM Tris–HCl pH 6.8, 1.5 mM EDTA, 20% glycerol, 2% SDS, 5% β-mercaptoethanol, 0.0025% bromophenol blue) after washing with PBS. Protein extracts were quantified, and equivalent amounts of extracts were resolved by SDS–polyacrylamide gel electrophoresis (PAGE) and electrotransferred to nitrocellulose membrane (Amersham, UK) [22]. The membrane was first blocked for 2 h with 5% skim milk in 1x TBS and incubated overnight at 4 °C with primary antibodies anti-filaggrin (1:1000, Genetex, CA, USA), anti-loricrin (1:1000, Genetex, CA, USA) in blocking buffer. The following day, the membrane was incubated with anti-rabbit IgG-HRP-linked (1:5000, Cell Signaling, MA, USA) for all targets after washes with TBS-tween 0.1%. Bands were detected using Immobilon Western Chemiluminescent HRP (Millipore, MA, USA) and detected by means of Imager instrument (Amersham, UK). Quantification of signal optical density was performed by ImageJ software.

### 2.13. In Vitro Assay

The effect of PFP application on vaginal mucosa was addressed by measuring cell viability in an in vitro model of human vaginal mucosa as previously described in [23,24,25,26,27]. Human vulvar carcinoma cells A431 were grown on the surface area (0.50 cm^2^) of a polycarbonate filter insert to obtain a cellular multilayer that histologically resembled and reconstructed the vaginal mucosa epithelium. (1) To evaluate the irritating capacity, the cellular multilayer was exposed to PFP aqueous solution at 0.1% (*w*/*v*) and 0.01% (*w*/*v*) for 24 h, then washed with saline solution at room temperature to eliminate the excess of treatment. Subsequently, cell viability was assessed by CCK-8 test [28]. As a negative control, the insert treated with saline solution was used, while the treatment with 0.5% sodium dodecyl sulfate (SDS) in aqueous solution was used as the positive control. (2) To evaluate the soothing effect of PFP, mucosal inserts, except for the reference ones, were exposed to a lasting irritant treatment (0.4% lactic acid in aqueous solution) for 1 h. After washing, the cellular multilayer was exposed to 100 µL of PFP solution 0.1%. The irritant control insert did not receive treatment, while acetylsalicylic acid solution 0.03% aqueous solution (ASP) was used as soothing agent control. Subsequently cell viability was assessed by CCK-8 test [28]. The TEWL (Transepidermal Water Loss) was also measured by calculating the evaporation rate (g/m^2^/h) by VapoMeter, Delfin. All tests were performed in duplicate. The in vitro assay was conducted by Eurochem Ricerche Srl laboratory (Viale del Lavoro, 6-35035 Mestrino PD—Italy), certified according to ISO 9001:2015. 

### 2.14. In Vivo Tests

The evaluation of the tolerability and effectiveness of the PFP treatment was performed by the topical application of two preparations composed of a cream and a mask with the following composition:

Cream: 3% *w/w* of PFP dispersed in vegetable glycerin (the content of PFP in the final formulation was 0.1% *w*/*w)*. The INCI of the cream was the following: Aqua, Propanediol, Glycerin, Arachidyl Alcohol, Butyrospermum Parkii Butter, Isoamyl Laurate, Coco-Caprylate, Cocoglycerides, Prunus Amygdalus Dulcis Oil, Caprylic/Capric Triglyceride, Behenyl Alcohol, Cetearyl Alcohol, Hydrogenated Vegetable Oil, Perilla Frutescens Callus Lysate, Arachidyl Glucoside, Hydroxyacetophenone, Xantham Gum, Glucose, Undecane, Tridecane, Citric Acid, Dimethicone, 1,2-Hexanediol, Caprylyl Glycol, Sodium Hydroxide, Ammonium Acryloyldimethyltaurate/Vp Copolymer. The placebo cream contained the same ingredients as just listed except the PFP.

Mask: 3% *w/w* of PFP dispersed in vegetable glycerin (the content of PFP in the final formulation was 0.1% *w*/*w*). The INCI of the mask was the following: Aqua, Glycerin, Caprylic/Capric Triglyceride, Olus Oil, Inulin, Perilla Frutescens Callus Lysate, Alpha-Glucan, oligosaccharide, Sodium Hyaluronate, Lauryl Glucoside, Polyglyceryl-2 Dipolyhydroxystearate, Dipotassium Glycyrrhizate, Arginine, Glyceryl oleate, Lactic Acid, Citric Acid, C18-22 Hydroxyalkyl Hydroxypropyl guar, Dicaprylyl carbonate, Caprylyl Glycol, Hydroxyacetonephenone, Dipropilene Glycol, Benzyl Alcohol, Dehydroacetic Acid. The placebo mask contained the same ingredients as just listed except the PFP.

The test was conducted according to the Helsinki Declaration on 30 female volunteers with an average age of 58.1. Subject selection was carried out according to the following criteria: Caucasian subject; female subjects; 40–70 years old; in overall good health; women in pre/post menopause, with vaginal dryness, irritation, and/or loss of tone; subjects able to follow all study direction and to commit to all follow-up visits for the entire study duration; and subjects who have completed the information process and signed the informed consent. The exclusion criteria of the subjects were the following: pregnant or nursing women; subjects with previous episodes of skin hypersensitivity to cosmetic products or who are sensitive to any of the test article components of the product to be tested; subjects who are taking topical or systemic drugs that could affect the test results (anti-inflammatory agents, corticosteroids, etc.); subjects with systemic diseases or skin disorders that may affect the outcome of the test or increase the risk for the volunteer; subjects who participated in clinical trials with comparable purposes in the 30 days before the start of the test; subjects with vaginal infections such as *Trichomonas* spp., *Candida* spp. and/or bacterial vaginosis; subjects with breast cancer, uterine cancer, or estrogen-dependent tumors; severe primary disease of kidneys and hematopoietic system; recent malignant tumors. During the study, the volunteers used the usual products for the daily cleaning of intimate areas, and they did not use the test products on areas different from those prescribed. For the entire duration of the study, subjects were not allowed to use different products on the testing area [29]. The recruited volunteer women were divided into two groups: 15 received the treatment (verum), and 15 the placebo.

The volunteers used the treatment (verum or placebo) consisting of cream and mask for 2 weeks, after cleaning the intimate area, in the following way:soothing, intimate cream: once a day by applying abundantly in the anogenital and surrounding area;mask: 3 times a week with an application of 20 min, making sure to use a protective system to avoid getting the garment wet.

During the treatment with the mask, subjects also used the cream. At the initial check-up and after 15 days of treatment, clinical and instrumental evaluations were performed. The gynecologist examined the external genitalia (vulva) and the perineal area, giving a score to the parameters of redness, dryness, peeling, swelling, blisters and the presence of secretions. The volunteers also reported any sensations of itching, burning, vaginal discharge, or other to the gynecologist. The intensity of the reactions was scored according to the four-point scale: 0 = absent, 1 = mild, 2 = moderate and 3 = severe. In correspondence with the treated area, non-invasive instrumental measurements of deep hydration and skin elasticity were made. The in vivo tests were conducted by Eurochem Ricerche Srl laboratory (Viale del Lavoro, 6-35035 Mestrino PD—Italy), certified according to ISO 9001:2015 and following the Guidelines for cosmetic ingredients testing by Scientific Committee on Consumer Safety [30,31,32].

#### Instruments and Parameters

The tolerability and efficacy of the treatments were assessed by a clinical examination of the external genitalia and the perianal area by the gynecologist. The instrumental efficacy of the treatments is highlighted by statistically significant improvements in the following cutaneous parameters investigated: deep hydration (evaluated by MoistureMeterEpid, Delfin Technologies) and elasticity (evaluated by Cutometer MPA 580, Courage & Khazaka). Moisture Meter Epid converts the measured dielectric constant into a percentage of water contained in the tissue up to a depth of 1 mm, which is proportional to skin hydration [33,34]. The percentage water content of the tissue is calculated as follows:Water% = [(ε − 1)/(ε_w_ − 1)] ∗ 100%
where ε is the dielectric constant measured, and ε_w_ is the dielectric constant of water (79.0). 

The Cutometer MPA 580 measures the vertical deformation of the cutaneous surface when it is aspirated into a measuring probe [33,35]. Data provided by the device are expressed in mm. In the final calculation of results, the following parameters are considered: skin extensibility, passive behavior of the skin subjected to a force (skin tone) and total elasticity of the skin including viscous deformation (it is the relationship between “recovery ability” and “skin extensibility”).

### 2.15. Statistical Analysis

In vitro tests: statistical analyses were performed using unpaired Student t-test or one-way analysis of variance (ANOVA) followed by Tukey post-hoc test (with *p* < 0.05 significance level) as appropriate. Data are presented as mean ± S.D. Analyses were conducted using Graphpad Prism 8.0 (San Diego, CA, USA). 

In vivo tests: Data are presented as mean ± S.D. Based on the results of normality test (*Kolmogorov*–*Smirnov*), the following statistical comparisons were performed:instrumental data (T_0_ vs. T_2weeks_) of both treatments were statistically compared by *Student t*-test for dependent and parametric data;clinical data (T_0_ vs. T_2weeks_) of both treatments were statistically compared by *Wilcoxon* test for dependent and non-parametric data;statistical comparisons between the active and placebo group for instrumental data were performed by *Student t*-test for independent and parametric data;statistical comparisons between active and placebo groups for clinical data were performed by *U Test of Mann–Whitney* for independent and non-parametric data.

The data groups were considered statistically different for a probability value of ≤0.05. It was not possible to apply any statistical test to the clinical parameters in which all the volunteers showed values equal to zero at both control times [30,31,36,37,38].

## 3. Results

### 3.1. Perilla Frutescens Phytocomplex Obtaining from a Selected Cell Line and Chemical Analysis

A stable cell line of *P. frutescens* was obtained using the perilla solid medium (medium Gamborg B5 supplemented with 20 g/L sucrose, 0.9% (*w*/*v*) of Plant Agar, 0.5 mg/L of NAA, 1 mg/L of IAA at pH 6.5). After six months in this selected solid medium, the *P. frutescens* cell line appeared pale, purple coloured, with a friable texture and a high growth rate (subculture in fresh solid medium every 21 days) (Figure 1a). The staining with fluorescein diacetate displayed the cell morphology and viability of the plant cells maintained in a solid perilla medium (Figure 1b,c). The content of RA, total polyphenols and anthocyanidins in the cell line was optimized using a perilla final liquid culture medium with a higher content of sucrose (50 g/L) and a lower concentration of growth hormones (NAA 0.3 mg/L and IAA 0.8 mg/L). The cells grown into perilla final liquid culture medium were used to prepare the PFP. To measure the content of RA, total polyphenols and anthocyanidins in PFP, UPLC-DAD analysis was performed. The UPLC-DAD chromatogram recorded at 330 nm is shown in Figure 2. The content of total polyphenols, identified by their characteristic spectra and expressed as equivalent of RA, was 2.35 ± 0.16% *w/w*; the content of RA, calculated measuring the peak area at retention time 7.5, was 2.03± 0.16% *w/w*. As observed in natural growth, the biotechnological culture of *P. frutescens* contained RA as the main polyphenol in the phytocomplex. The content of total anthocyanidins identified by their characteristic spectrum with λ_max_ at 520 nm and expressed as equivalent of cyanidine-3-O-glucoside was 0.10 ± 0.02% *w/w*.

### 3.2. PFP Effect on the Viability of Human Keratinocytes and Fibroblasts

The first aim of the study concerned the evaluation of the effect of PFP on the viability of keratinocytes and fibroblasts, the main cell populations composing the epidermis and dermis, respectively, whose interaction is fundamental for skin homeostasis and regeneration. HaCaT and HFF cells were treated with different concentrations of PFP for 24 h, and cytotoxicity was investigated through the measurement of cellular metabolism by using the CCK-8 assay. The statistical analysis indicated that PFP at the lower dosage (0.1 µg/mL) was able to improve significantly the viability of both HaCaT and HFF cells compared to control ones (one-way ANOVA: HaCaT, *p* = 0.0025; HFF, *p* = 0.0095 vs. CTRL) (Figure 3a,b). In contrast, treatment with PFP 1000 µg/mL significantly decreased cell viability in both HaCaT and HFF cells (one-way ANOVA: HaCaT: *p* = 0.0063; HFF: *p* < 0.0001 vs. CTRL) (Figure 3a,b). Based on these data, the subsequent analyses were carried out using the no cytotoxic concentrations (1, 10, or 100 µg/mL).

### 3.3. PFP Does Not Stimulate Pro-Collagen I and Elastin Production in HFF but Enhances the Keratinocytes Migration Ability

Re-epithelialization of skin wounds involves an orderly series of events in which fibroblasts produce and organize the extracellular matrix (ECM) components to re-form the basement membrane beneath the epidermal basal layer, while keratinocytes migrate, proliferate and differentiate to restore the barrier function [39]. Collagen and elastin are the most abundant fibrous proteins composing ECM and are produced by fibroblasts. To investigate the possible activity of the PFP on skin repair, we first analyzed the effect of the PFP treatment on the ability of fibroblasts to produce collagen. By means of ELISA assay, the release of pro-collagen I and elastin was measured in HFF after treatment with PFP (1, 10, 100 µg/mL) for 24 h. No differences were observed in pro-collagen I and elastin protein levels between treated and control cells.

Secondly, we evaluated the impact of the PFP on the keratinocytes migration capacity in a wound-healing in vitro model. An artificial scratch was created on the cell confluent monolayer, and the ability of the keratinocytes to close the wound was monitored by measuring the wound area at 6 h and 24 h after the treatment with PFP (1, 10, 100 µg/mL). The analysis of the scratch wound healing assay revealed a main effect of the phytocomplex (100 µg/mL) in stimulating HaCaT migration after 6 h and more after 24 h compared to control cells (Figure 4a,b). Indeed, although not statistically significant, the increase in the healing rate of the keratinocyte monolayer produced by PFP (100 µg/mL) at 6 h was about 35% compared to the control (Figure 4a). After 24 h, the healing rate significantly improved by over 40% (one-way ANOVA: *p* = 0.0452 vs. CTRL) (Figure 4b). At the lower concentrations, the PFP showed no activity, neither at 6 h nor at 24 h.

### 3.4. PFP Counteracts Keratinocytes Inflammatory Response by Reducing Pro-Inflammatory Cytokines Release 

With the aim of evaluating the effect of PFP in a condition of inflamed skin, we assessed different in vitro models of inflammatory response. 

First, we assessed an in vitro model of inflamed epidermis by stimulating keratinocytes with factors released by active infiltrating monocytes. According to the cell viability test, HaCaT cells were treated for 24 h with PFP at the non-toxic concentrations (1, 10, 100 µg/mL) and simultaneously exposed to LCTM. To ensure the effectiveness of the inflammatory stimulus, protein levels of the pro-inflammatory cytokines TNF-α, IL-6, and IL-1β were measured by using ELISA assay in treated HaCaT cells. After 24 h of exposure, TNF-α resulted in being the most upregulated cytokine with a significant increase of its levels by 10-fold compared to the control group (one-way ANOVA: *p* < 0.0001 vs. CTRL) (Figure 5a). IL-1β and IL-6 production significantly increased compared to the unstimulated cells (one-way ANOVA: IL-1β *p* = 0.0003, IL-6 *p* = 0.0148 vs. CTRL) (Figure 5a). The treatment with PFP was able to counteract the high levels of TNF-α, IL-6, and IL-1β induced by LCTM. Indeed, statistical analysis revealed that, in cells exposed to the conditioned medium, PFP treatment at the concentration of 100 µg/mL prevented the increase of TNF-α, IL-1β, and IL-6 levels (*post hoc*: TNF-α *p* < 0.0001, IL-1β *p* = 0.0026, IL-6 *p* = 0.0468, vs. LCTM) (Figure 5a). At the concentrations of 1 and 10 µg/mL, PFP turned out to be ineffective. 

The quantification of TNF-α, the most regulated cytokine, was evaluated in HaCaT cells treated with PFP (100 µg/mL) and exposed to the endotoxin LPS (1 µg/mL) for 24 h. The analysis showed a significant increase of TNF-α levels in HaCaT cells stimulated with LPS compared to controls (one-way ANOVA: *p* = 0.0091 vs. CTRL) (Figure 5b). The treatment with PFP alone did not impact TNF-α production, while in LPS-stimulated cells, it counteracted the increase of LPS–induced increase of TNF-α levels (*post hoc*: *p* < 0.05 vs. LPS) (Figure 5b). 

We further investigated the effect of PFP in modulating cytokines release by evaluating the immune response of active monocytes. THP-1 cells were treated with PFP (1, 10, 100 µg/mL) concomitantly to LPS (200 ng/mL) for 24 h. A significant increase in TNF-α levels was observed in THP-1 cells stimulated by LPS compared to control cells (one-way ANOVA: *p* < 0.0001 vs. CTRL) (Figure 5c). PFP treatment was not effective in regulating the expression levels of TNF-α, both in unstimulated and LPS-stimulated cells (Figure 5c).

These data showed that PFP was able to inhibit the pro-inflammatory cytokines release specifically in keratinocytes indicating the potential activity of the biotechnological PFP in skin inflammatory disorders.

### 3.5. PFP Inhibits JNK Activation in HaCaT Cells Exposed to LPS-Conditioned THP-1 Medium

The mitogen-activated protein kinases (MAPKs) signaling pathway represent the major intracellular mechanisms involved in the regulation of the cellular responses to external stress signals and are crucial for the downstream production of pro-inflammatory mediators. MAPKs signaling plays a fundamental role in inflammation-based disorders and diseases at the skin level [40,41,42,43]. Among the MAPKs, the c-Jun N-terminal kinases (JNK) pathway has been reported to be involved in the barrier function of the epidermis [44,45,46]. Given the peculiar ability of PFP in modulating cytokines release induced by a 24 h inflammatory stimulus, we investigated whether this effect could be associated with an early regulation of MAPKs activity by measuring the phosphorylation levels of JNK in HaCaT cells stimulated with LCTM for 60 min through the ELISA dosage. As previously described [20], JNK activation peaked at 60 min with a significant increase of about 2-fold compared to control cells (one-way ANOVA: *p* = 0.0022 vs. CTRL) (Figure 6). Statistical analysis revealed that treatment with PFP, at the dose of 100 µg/mL, was effective in counteracting significantly the LCTM-induced increased levels of JNK phosphorylation after 60 min (post hoc: *p* = 0.0019 vs. LCTM) (Figure 6). Despite not being statistically significant, PFP alone was able to reduce the phosphorylation levels of JNK.

In light of these results, we suggest that the activity of PFP on keratinocyte inflammatory response could be mediated by the phosphorylation of JNK MAPK.

### 3.6. PFP Treatment Reduces the Leukocytes Infiltration Induced by Immune Stimulus

Inflammatory condition alters the permeability and integrity of the epidermis, compromising the barrier function. For the resolution of the inflammation, the recruitment of leukocytes in the site of damage and the leukocyte-derived inflammatory mediators is crucial; indeed, circulating leucocytes reach the inflamed epidermis transmigrating through the endothelial cells.

For this reason, we evaluated the effect of a pre-treatment with PFP (100 µg/mL) in an in vitro model of epidermis inflammation obtained by a co-culture of LPS-stimulated HaCaT cells and THP-1 monocytes.

The cytological staining with haematoxylin-eosin showed the monocytes adhesion by creating infiltration zones (blue spot) and determining the alteration of the HaCaT monolayer exposed to LPS compared to unstimulated cells (Figure 7a,b). The pre-treatment with the effective concentration of PFP prevented the alteration of the keratinocytes monolayer reducing the monocyte infiltration and the inflammatory damage (Figure 7c). This data corroborated the effect of PFP in reducing the inflammatory response, specifically in keratinocytes cells.

### 3.7. PFP Prevents Keratinocytes Tight Junctions Impairment Induced by an Inflammatory-Oxidative Damage

One of the most relevant aspects in evaluating epidermis integrity concerns the analysis of tight junctions (TJs). TJs consist of transmembrane multiprotein complexes, such as occludin, claudins and adhesion molecules that are linked to the cytoskeleton by zonula occludens (ZO) proteins, which serve as regulatory proteins to the TJs [47,48]. These proteins interconnect the cells of the granular layer, ensuring the barrier function in healthy epidermis. 

Moreover, the skin barrier is guaranteed by two important proteins such as filaggrin and loricrin. Filaggrin aggregates keratin filaments and promotes cytoskeleton condensation and cell compaction to form the cornified envelope [49,50]. Loricrin is one of the most important structural proteins expressed in the granular layer that contributes to the formation of the cornified layer [49,51].

These premises led us to investigate protein expression of occludin and ZO-1 by immunostaining in HaCaT cells pre-treated with PFP (100 µg/mL) for 2 h and then exposed to LPS+H_2_O_2_ for 3 h. The inflammatory/oxidative stimulus was previously optimized to obtain an evident TJs impairment in HaCaT cells. Immunofluorescence images displayed the expression of occludin and ZO-1 in control and PFP-treated cells (Figure 8). The exposure to LPS+H_2_O_2_ for 3 h affected the ZO-1 junction expression showing a fragmented distribution along the cell membrane of HaCaT cells compared to the control (Figure 8a). 

The 2 h pre-treatment with PFP was able to prevent LPS+H_2_O_2_-induced membrane damage by preserving the ZO-1 expression (Figure 8a). Similarly, the expression of the TJ occludin was severely reduced in the HaCaT cell membrane exposed to LPS+H_2_O_2_, whereas PFP pre-treatment was able to completely inhibit the occludin junction impairment promoted by the inflammatory-oxidative stimulus (Figure 8b). Employing western blot, we analyzed the protein expression of filaggrin and loricrin in HaCaT cells in the same condition. Interestingly, the stimulation of keratinocytes with LPS+H_2_O_2_ for 3 h significantly increased filaggrin and loricrin protein levels compared to the unstimulated counterpart (one-way ANOVA: *p* < 0.0001 vs. CTRL). Statistical analysis showed that PFP at 100 µg/mL did not affect filaggrin (Figure 9a) and loricrin protein levels (Figure 9b) compared to control cells but repressed the LPS+H_2_O_2_-induced increased levels of both filaggrin and loricrin (*post hoc:* filaggrin *p* = 0.0467, loricrin *p* = 0.0203 vs. LPS+H_2_O_2_) (Figure 9).

These data revealed that PFP was able to maintain the integrity of the skin barrier by regulating the expression levels of TJs, and filaggrin and loricrin proteins, two crucial components of the skin barrier integrity.

### 3.8. PFP Solution Shows Non-Irritant and Soothing Properties in an In Vitro Reconstituted Vaginal Mucosa

Driven by very positive results obtained investigating the effect of PFP at the cellular and molecular level, we aimed to study PFP in a practical context of a functional application by assessing the irritant and soothing potential of PFP in an in vitro model that resembles the vaginal mucosa. 

The reconstruction of vaginal mucosa was obtained by a multilayer of human vulvar carcinoma (A430) cells grown on the top of an inert support. To evaluate the possible irritant effect of PFP, cell viability was analyzed on the cell multilayer treated for 24 h with an aqueous solution of 0.5% sodium dodecyl sulfate (SDS) as irritation positive control, a saline solution as the negative control, and an aqueous solution of PFP at the concentration of 0.1% and 0.01%. A marked effect of the SDS solution was showed impairing cell viability by about 90% with respect to physiological control. In contrast, PFP aqueous solution at 0.1% and 0.01% did not affect the cell viability of reconstituted mucosa, similarly to the control sample (Figure 10a). Moreover, we analyzed the soothing ability of PFP in an irritation condition by using the MTT test; the vaginal mucosa model was treated with an aqueous solution of 0.4% lactic acid (LA) for 1 h, then exposed to 0.1% PFP solution and 0.03% acetylsalicylic acid solution (ASP) for 4 h, as soothing agent control. 

Results displayed that LA solution affected cell viability by about 15% compared to untreated mucosa. The treatment with the ASP was able to counteract the harmful effect induced by LA, enhancing cell viability by about 10%. Similarly, although to a lesser extent, PFP 0.1% solution was able to ameliorate the cell viability of the vaginal mucosa compared to LA treatment (Figure 10b). To further assess the effect of PFP on skin hydration, the quantity of condensed water that diffuses across a fixed area of stratum corneum was also measured by using TEWL. The reconstituted vaginal mucosa was exposed to the irritant agent, 0.4% LA, for 1 h, then treated with 0.1% PFP or 0.03% ASP solution for 4 h. After treatment with LA, TEWL was markedly increased, while the PFP application reduced the TEWL at a comparable level to that of the ASP (Figure 10c). 

### 3.9. PFP Ameliorates Deep Hydration and Elasticity after In Vivo Topical Application

In light of the previous evidence, we aimed to assess the tolerability, effectiveness and safety of two topical preparations, a mask and a cream, containing the active PFP in an in vivo application. For the study, 30 female volunteers, with an average age of 58.1 years, were recruited and divided into two groups receiving the active PFP preparations or the placebo, respectively, for two weeks. At the end of the treatment, the gynecologist examination was performed by assigning a score to the parameters of redness, dryness, peeling, swelling, blisters and presence of secretions according to an established scale. The statistical analysis reported a significant increase in deep hydration in women treated with PFP compared to the placebo counterpart. Among the placebo group, no statistically significant changes were recorded (Figure 11) (Table 1).

Concerning skin elasticity, after two weeks, no significant differences were observed in the value of skin extensibility between PFP- and placebo-treated women (Figure 11). In contrast, the treatment with the active PFP improved the total elasticity of the skin compared to the placebo group. No statistically significant changes were measured in the placebo group (Figure 11) (Table 2).

Moreover, a clinical evaluation of the effectiveness and compatibility of vaginal dryness and irritation in pre/post-menopausal women was carried out. The PFP treatment decreased redness and drying in the perianal and external genitalia areas compared to the placebo treatment, as well as the swelling value of external genitalia. The collection of these data allowed us to evaluate the effective efficacy and tolerability of PFP in vivo, directly on an application area such as the vaginal mucosa, by using simple formulations such as a cream and a mask.

## 4. Discussion

The use of in vitro plant-derived cell cultures is arising as a sustainable and standardized way to obtain sources of active secondary metabolites. This is particularly important when botanical species need attention for their preservation. Still, it is also worth considering when the quality standards of products available on the market are far from achieved. This study proposed an innovative *Perilla frutescens* (L.) Britton product, a biotechnological phytocomplex (PFP) characterized by a standardized phytochemical profile enriched in polyphenols and, in particular, RA, to whom many biological properties of perilla are referred to the field of skin protection from inflammatory dysregulations [4,5]. Skin barrier impairment may cause penetration of external antigens resulting in the onset of inflammation conditions. Filaggrin and loricrin are crucial epidermal barrier proteins, and their expression is correlated to the levels of pro-inflammatory cytokines, such as TNF-α [50]. Besides these, also TJs guarantee an efficient barrier function contributing to correct intercellular adhesion. Dysfunction of TJs associated proteins may result in the disruption of barrier integrity and disorders in the epithelial layer [47,48]. Based on this knowledge, RA-enriched PFP was tested for its anti-inflammatory activity and ability to preserve skin barrier function in human keratinocytes in in vitro models [13,52]. PFP exerted sounding effectiveness at 100 µg/mL in maintaining the integrity of the skin epithelial barrier through the regulation of both skin barrier proteins (loricrin and filaggrin) and TJs (occludin and ZO-1) expression altered in an inflammatory and oxidative stress condition. The results were supported by the ability of PFP in promoting re-epithelialization in a wound healing assay. Previous papers reported the beneficial effect of *P. frutescens* on the intestinal epithelial barrier by targeting intestinal permeability [53], whereas Ye-Ram Kim et al. showed the increased proliferation and migration of HaCaT cells treated with isoegomaketone, a phytochemical isolated from *P. frutescens* leaves [54]. PFP was not as effective in modulating human fibroblast metabolism as it was in human keratinocytes, and it was devoid of capacity in modulating pro-collagen I and elastin synthesis. As regards epidermis inflammation, we chose to test PFP in a co-culture of HaCaT and THP-1 cells that resulted in a dynamic and more realistic inflammatory model, where monocytes participate in the infiltration process associated with modified integrity of inflamed skin epithelium. This model allowed us to find that PFP exerted an anti-inflammatory activity by inhibiting the release of pro-inflammatory cytokines TNF-α, IL-1β and IL-6 in LPS-stimulated monocytes conditioned keratinocytes, according to evidence on the activity of conventional preparations from *P. frutescens* containing RA [55]. Interestingly, we demonstrated a specific anti-inflammatory activity of PFP on human keratinocytes by separately dosing cytokine levels in LPS-stimulated HaCaT and THP-1 cells. As a consequence of the difficulty in studying standardized preparations, in the last 20 years, many papers mainly focused on isolated components of perilla to explore the potentiality of the species as a skin protective agent [55], but they did not address the peculiar biological effects of perilla phytocomplex. Indeed, RA is the major polyphenol also in *Melissa officinalis* L. leaves, for example, but the herbal preparations from this species did not demonstrate a clear skin anti-inflammatory effect; in the case of *Rosmarinus officinalis* L. leaves, enriched in RA as well, the skin protective activity is mainly related to the presence of carnosic acid and carnosol [56]. This study finally demonstrated the peculiar activity of the phytocomplex of perilla in modulating skin inflammatory response. In this work, we also elucidated the underlying anti-inflammatory mechanism of PFP in human keratinocytes. Phosphorylation of MAPKs is one of the main upstream intracellular signaling pathways associated with downstream production of pro-inflammatory cytokines and mediators. We examined the JNK pathway, a subfamily of MAPKs, which is implicated in many immune-related skin disorders regulating a wide range of cellular processes, including cell proliferation, differentiation, survival, apoptosis and inflammation [44]. PFP targeted the MAPK pathway by reducing JNK activation in stimulated HaCaT cells. A related mechanism has been described for a not chemically characterized methanolic extract of *P. frutescens* in mouse macrophages: it reduced NO production, PGE2 secretion and pro-inflammatory cytokines (TNF-α and IL-6) in LPS-stimulated macrophages by downregulating the mRNA expression and protein production of pro-inflammatory mediators and inhibiting the ERK1/2, JNK, p38, as well as NF-κB signaling [57,58,59]. We could state that PFP is a noticeable example of how phytotherapy, even when innovative such as in biotechnology, actually has a major role in inflammatory skin impairments. Indeed, the ability of some phytocomplexes to modulate immune pathways with a multitarget mechanism is well-known and exploited in the pharmaceutical sector, as reported in 32 specific monographs of the European Medicine Agency (EMA) [60] and in all four volumes of World Health Organization (WHO) monographs on selected medicinal plants [61]. Herbal products also are very well used in functional cosmetics to maintain skin homeostasis and prevent dysregulation physiologically caused by environmental factors or age [62]. With the aim of transferring PFP in a practical healthcare framework, we believed that it could be rationally thought of as an ingredient for intimate care, particularly in pre- and post-menopausal women, in which the drop in circulating hormones levels, especially oestrogen, represents the main trigger of vaginal mucosa atrophy. Thinning and dryness of the vaginal epithelium increase susceptibility to trauma, resulting in flattened epithelial surfaces, features of keratinization, and absence of papillae, as well as the higher vulnerability of the mucosa to inflammation or infection [63,64]. Mungmai et al. [65] showed the efficacy of a cosmetic formulation (serum) containing a non-characterized *P. frutescens* leaf extract (PLE) on irritation and aged skin. Clinical evaluation conducted on healthy volunteers indicated that the serum containing PLE could increase skin hydration and skin elasticity after treatment. To support the idea of functional cosmetic usage, we performed validated tests in vitro on a reconstructed vaginal mucosa model that highlighted the non-irritant and soothing properties of PFP (0.1%) and its ability to reduce the TEWL effect after treatment with the irritant agent (0.4% lactic acid in aqueous solution). These data were corroborated in a very preliminary but fundamental in vivo evaluation performed according to approved cosmetic tests on healthy volunteers. The application of two topical preparations containing PFP, 0.1% *w/w*, was tested in vivo by clinical and instrumental evaluation of tolerability and effectiveness of the treatment in the intimate area. Topical treatment for two weeks with mask and cream containing PFP improved the integrity of vaginal mucosa among the women in pre- and post-menopause. A trend toward improved deep skin hydration and total elasticity was statistically significant against a placebo. Skin hydration and elasticity are strictly related to skin barrier proteins and TJs integrity, and we could postulate that in vivo findings correlate with epidermis protective mechanisms that we showed for PFP in the in vitro cell model. Overall, data allow us to define the strengths and limits of this work: PFP is an innovative biotechnological phytocomplex derived from *P. frutescens*, characterized by a standardized chemical composition where rosmarinic acid content is higher than in naturally growing perilla; PFP has sound effectiveness in protecting epidermis integrity in inflammatory condition by downregulating inflammatory response through JNK MAPK modulation and by normalizing TJs and barrier proteins levels; PFP is suitable for cosmetic applications, and it improves vaginal hydration in vitro and, when used in a topical formulation in vivo, ameliorates elasticity and hydration in intimate care. On the other hand, PFP effectiveness at the dermis level and the protective effect on fibroblasts need further investigation. Moreover, larger clinical evaluations are needed to ensure the healthy properties of PFP.

## 5. Conclusions

*Perilla frutescens* (L.) Britton biotechnological phytocomplex produced by in vitro plant cell culture technology, enriched in polyphenols, and in particular rosmarinic acid, proved to be an innovative ingredient with a high profile of safety and effectiveness thanks to its standardized, sustainable and clean production process. The in vitro study in the human keratinocytes model showed the anti-inflammatory and protective activities of PFP on the skin barrier. The innovative PFP also highlighted hydrating and soothing activities in the 3D reconstructed vaginal mucosa and improved the integrity of vaginal mucosa by increasing deep hydration and total elasticity in clinical tests. 

Overall, our findings highlight that PFP could be an excellent ingredient for cosmetic preparations for topical use, such as in intimate care, where it has shown good activity in preserving the integrity of the vaginal mucosa, especially in vulnerable conditions such as during menopause. Moreover, *P. frutescens* could have potential application as a topical agent in dermatological disorders that involve an inflammatory condition and compromised skin integrity.

## 6. Patent

ITA1020200000028230. International Application Number: WO 2022/112862. Phytocomplex and extract of a meristematic cell line selected from *Perilla frutescens*.

## Figures and Tables

**Figure 1 pharmaceutics-15-00240-f001:**
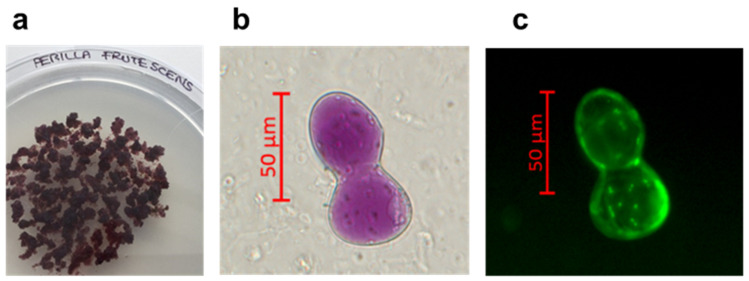
*Perilla frutescens* (L.) Britton cell culture maintained in solid perilla medium (**a**), optical images of *Perilla frutescens* cells observed by AXIO-Imager A2 optical microscope (ZEISS), in the bright field mode (**b**), and after staining with fluorescein diacetate (**c**). Scale bar: 50 µm.

**Figure 2 pharmaceutics-15-00240-f002:**
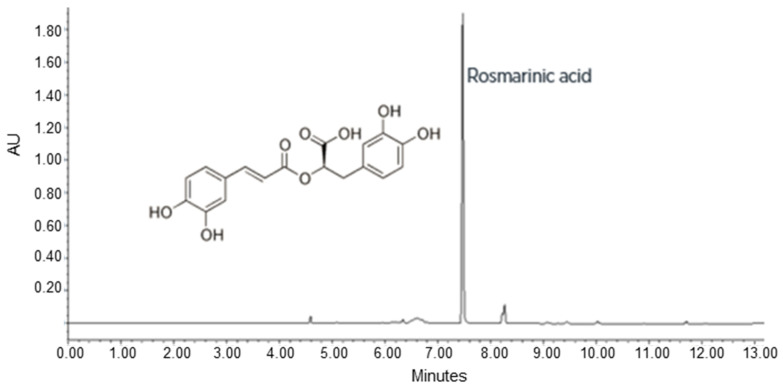
Representative UPLC chromatogram of *Perilla frutescens* (L.) Britton extract recorded at 330 nm. RA, which elutes at 7.5 min, represents the main component. RA—rosmarinic acid.

**Figure 3 pharmaceutics-15-00240-f003:**
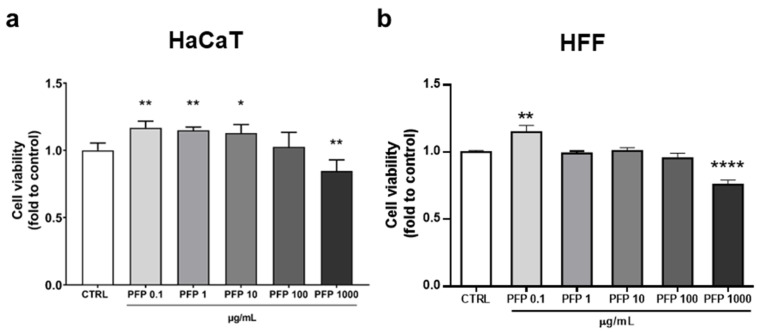
Cell viability analysis by CCK-8 assay on HaCaT (**a**) and HFF (**b**) cells treated with PFP at the concentrations of 0.1, 1, 10, 100 and 1000 µg/mL for 24 h. Each column represents mean  ±  SD. Data were analyzed by one-way analysis of variance followed by Tukey: * *p* < 0.05, ** *p* < 0.01, **** *p* < 0.0001 vs. CTRL.

**Figure 4 pharmaceutics-15-00240-f004:**
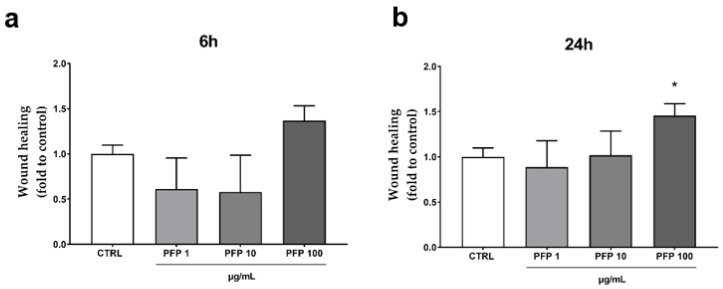
Wound Healing assay in HaCaT cells treated with PFP (1, 10, 100 µg/mL) for 6 h (**a**) and 24 h (**b**) or untreated (CTRL). Each column represents mean  ±  SD. Data were analyzed by one-way analysis of variance followed by Tukey: * *p* < 0.05 vs. CTRL.

**Figure 5 pharmaceutics-15-00240-f005:**
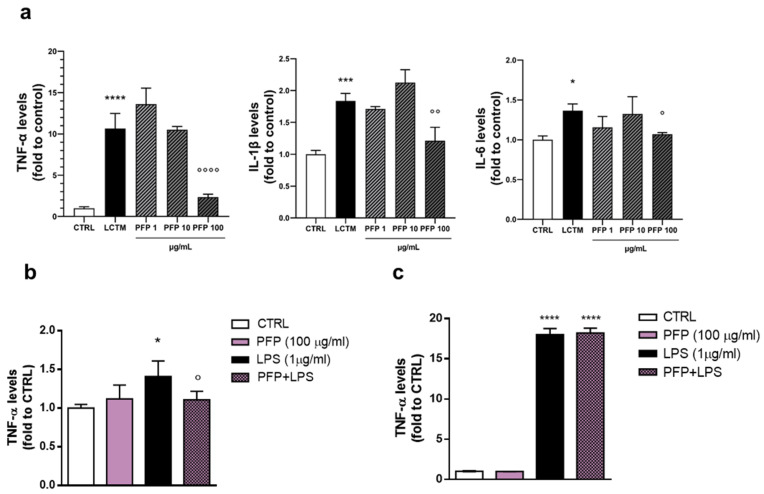
Effect of PFP (1, 10, 100 μg/mL) treatment on (**a**) TNF-α, IL-6, or IL-1β release analyzed by ELISA assay in HaCaT cells exposed to LPS-conditioned THP-1 medium (LCTM) for 24 h; (**b**) TNF-α in HaCaT cells stimulated by LPS (200 ng/mL) for 24 h; (**c**) TNF-α in THP-1 cells exposed to LPS 1 µg/mL) for 24h h. Each column represents mean  ±  SD. Data were analyzed by one-way analysis of variance followed by Tukey:* *p <* 0.05, *** *p* < 0.001,**** *p* < 0.0001 vs. CTRL; ° *p* < 0.05,°° *p <* 0.01, °°°° *p* < 0.001 vs. LCTM or LPS.

**Figure 6 pharmaceutics-15-00240-f006:**
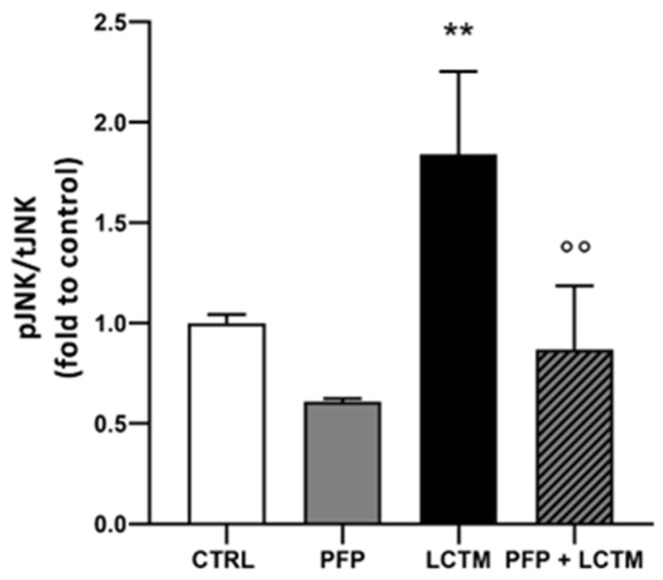
Effect of PFP (100 μg/mL) treatment on phosphorylation levels of JNK analyzed by ELISA assay in HaCaT cells unstimulated or exposed to LCTM for 60 min. Phospho-JNK levels (pJNK) were normalized on total JNK protein (tJNK). Each column represents mean  ±  SD. Data were analyzed by one-way analysis of variance followed by Tukey: ** *p* < 0.01 vs. CTRL; °° *p* < 0.01 vs. LCTM.

**Figure 7 pharmaceutics-15-00240-f007:**
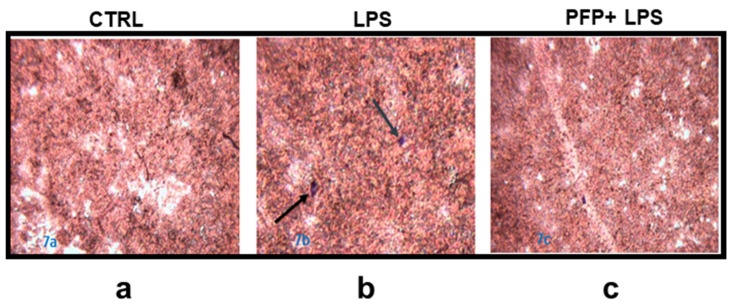
Microscope images of H&E staining of co-culture of THP-1 cells and HaCaT cells previously pre-treated with PFP (100 µg/mL) for 2 h and then stimulated with LPS for 24 h. (**a**) Untreated control HaCaT cells, (**b**) HaCaT cells stimulated with LPS, (**c**) HaCaT cells pre-treated with PFP and then stimulated with LPS.

**Figure 8 pharmaceutics-15-00240-f008:**
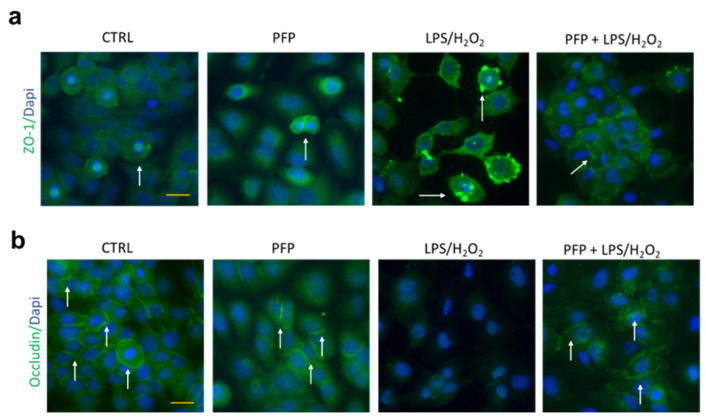
Immunofluorescence staining of (**a**) ZO-1 (green) and (**b**) occludin (green) TJs in HaCaT cells unstimulated (CTRL) or PFP-treated (100 µg/mL), or pre-treated with PFP for 2 h and exposed 3 h to LPS+H_2_O_2_. Dapi (blue) stained cell nuclei. Scale bar: 100 µm.

**Figure 9 pharmaceutics-15-00240-f009:**
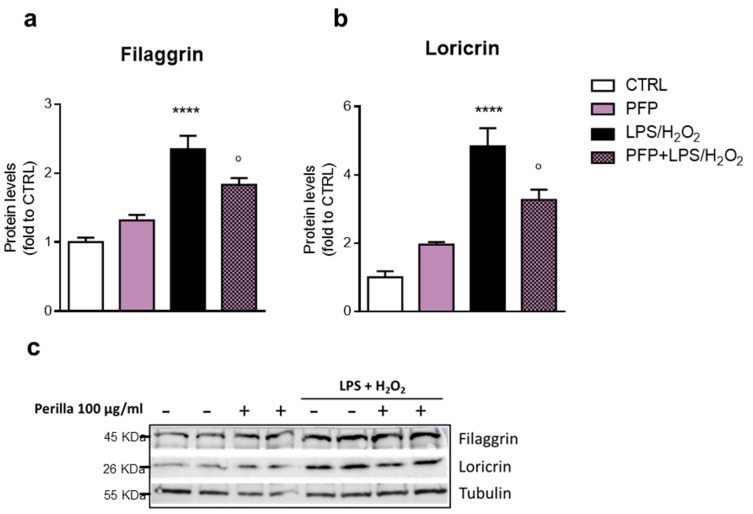
Western blot analysis of (**a**) filaggrin and (**b**) loricrin protein levels in HaCaT cells unstimulated (CTRL), or PFP-treated (100 µg/mL), or pre-treated with PFP for 2 h and exposed 3 h to LPS+H_2_O_2_. (**c**) Immunoblot represents protein levels of filaggrin and loricrin and the endogenous target tubulin. Each column represents mean  ±  SD. Data were analyzed by one-way analysis of variance followed by Tukey: **** *p* < 0.0001 vs. CTRL; ° *p* < 0.05 vs. LPS+H_2_O_2._

**Figure 10 pharmaceutics-15-00240-f010:**
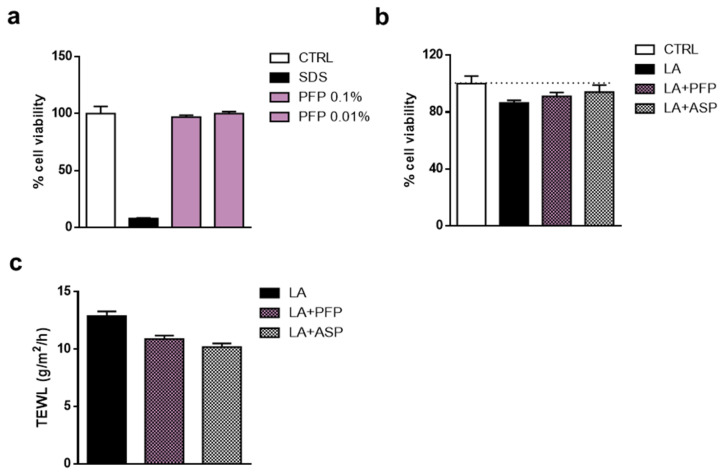
Analysis of (**a**) irritant, (**b**) soothing and (**c**) hydration activities of PFP 0.1% solution in an in vitro model of reconstructed vaginal mucosa. SDS 0.5% and lactic acid (LA) 0.4% solutions were used as irritant controls; acetylsalicylic acid (ASP) 0.03% solution was used as an active control agent. Data are expressed as mean ± SD of two independent experiments (*n* = 2).

**Figure 11 pharmaceutics-15-00240-f011:**
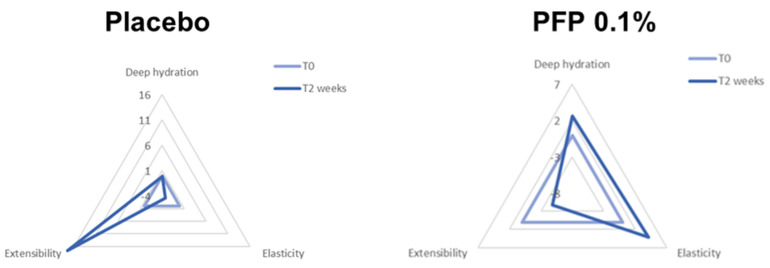
Observational analysis of deep hydration, elasticity and extensibility parameters of vaginal area of 30 female volunteers at starting time (T0) and after two weeks (T2) of treatment with PFP 0.1%-containing, or placebo, cream and mask. Data are expressed as the total score assigned following gynecologist evaluation.

**Table 1 pharmaceutics-15-00240-t001:** Score values obtained by deep hydration examination of active PFP-treated or placebo groups. Averages, standard deviations, variations and statistical significance (*p*-value) for each group and between groups are reported. * *p* < 0.05, ** *p* < 0.01.

	T0	T2 Weeks	Variation % T2 Weeks-T0	*p*-Value T0 vs. T2 Weeks
**PFP (0.1%)**	33.0 ± 4.9	35.6 ± 4.8	+7.9%	*p* < 0.01 **
**PLACEBO**	35.4 ± 4.6	35.3 ± 5.2	−0.1%	*p* > 0.05
**PFP vs. PLACEBO**				*p* = 0.01 *

**Table 2 pharmaceutics-15-00240-t002:** Score values obtained by elasticity examination of active PFP-treated or placebo groups. Averages, standard deviations, variations and statistical significance (*p*-value) for each group and between groups are reported. * *p* < 0.05.

	T0	T2 Weeks	Variation % T2 Weeks-T0	*p*-Value T0 vs. T2 Weeks
**PFP (0.1%)**	0.822 ± 0.084	0.856 ± 0.042	+4.1%	*p* = 0.052
**PLACEBO**	0.814 ± 0.063	0.788 ± 0.056	−0.1%	*p* > 0.05
**PFP vs. PLACEBO**				*p* < 0.05 *

## Data Availability

The authors do not have permission to share data.

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
