# Peer review of "A Novel Perilla frutescens (L.) Britton Cell-Derived Phytocomplex Regulates Keratinocytes Inflammatory Cascade and Barrier Function and Preserves Vaginal Mucosal Integrity In Vivo"

_pharmaceutics, 2023, doi:10.3390/pharmaceutics15010240_

Round 1

Reviewer 1 Report

The paper is well designed and performer.  However, the study is complex and difficult to read – thus they are some point need to be clarified:

1.     There are no specific tests on the skin barrier, for example, transdermal water loss. Therefore, the conclusion (also the title) is exaggerated and conclusions based on the results should be drawn with caution.

2.     In vivo study involved only vaginal mucosa, thus the sentence in the Conclusion „The clinical test performed in women in pre and post-menopause showed that topical application of PFP improved the  integrity of vaginal mucosa by increasing deep skin hydration and total elasticity.” (line 909/910) should be reworked in order to avoid the term „skin”.

3.     How Authors ensured sanitary safety of use the non-invasive measuring probes (hydration, elasticity) of external genitalia?

4.     The introduction part is too general and do not describe the aim of the work.

5.     There are some editorial errors (eg. description of figure 11 „observation” should be capital letter).

Reviewer 2 Report

This is an interesting paper that provides comprehensive evidence demonstrating the potential of a novel Perilla frutescens (L.) Britton cell-derived phytocomplex to exert anti-inflammatory and skin barrier protective activities, and preserve vaginal mucosal integrity in vivo. The authors have done a good job of presenting the results of the experiments and tying them to the proposed hypothesis. However, there are some areas that need further improvement before publication.

1.     Most of the sub section of the methodology is lacking standard reference. Please provide the same.

2.     Additionally, the authors should provide further evidence to prove that the tested phytocomplex has no toxic effects on animals.

3.     The authors should provide further discussion on the potential applications of the phytocomplex and its potential clinical implications.

4.     Text in the Figure 6, 8, 9, 10 is overlapping and not readable. Rectify it.

5.     Conclusion part is not up to the mark. Need to be redrafted.

6.     32% Plagiarism has been detected. Not in the acceptable range.

7.     Do you permission from ethical committee for conducting in vivo studies? Please provide permission details with reference number.

Round 2

Reviewer 2 Report

Necessary and suggested changes have been made in the revised manuscript and thus the manuscript can be accepted in its present form.